# Optimized Dynamic Monitoring and Quality Management System for Post-Harvest Matsutake of Different Preservation Packaging in Cold Chain

**DOI:** 10.3390/foods11172646

**Published:** 2022-08-31

**Authors:** Zihan Yang, Jinchao Xu, Lin Yang, Xiaoshuan Zhang

**Affiliations:** 1College of Engineering, China Agricultural University, Beijing 100083, China; 2Beijing Laboratory of Food Quality and Safety, College of Engineering, China Agricultural University, Beijing 100083, China; 3College of Food Science, Tibet Agricultural and Animal Husbandry College, Linzhi 860000, China

**Keywords:** Tibet matsutake, cold chain, monitoring system, preservation packaging, quality management

## Abstract

The quality of Tibetan matsutake drops during cold chain transportation. To extend the shelf life and improve the market value, this study analyzed the matsutake logistics process, and optimized the dynamic monitoring and quality management systems for post-harvest matsutake with different preservation packaging in the cold chain. This system monitored the micro-environmental parameters of the cold chain in real time, and it identified the best preservation method by analyzing the quality change characteristics of the matsutake with different preservation packaging. It was concluded that the matsutake were best preserved under the conditions of modified atmosphere packaging. The data analysis on the collected data verified the performance of the system. Relevant personnel were invited to participate in the system performance analysis and offer optimization suggestions to improve the applicability of the established monitoring system. The optimized model could provide a more effective theoretical reference for the dynamic monitoring and quality management of the system.

## 1. Introduction

The Tibetan matsutake is a precious, wild edible fungi, known as the “king of fungi”. It has high nutritional, medicinal and economic values, and is deeply appreciated by people [1,2]. However, due to the high water content of fresh matsutake mushrooms, with their tender tissue and no protective structure on the surface, they can easily rot in the cold chain transportation and storage after harvesting. In addition, compared with other edible fungi, the matsutake has a short fresh-keeping period. Generally, it will open umbrella in 2 days, then brown and start autolysis after it is stored at room temperature, losing its edible value, resulting in food waste and economic losses [3]. Due to the long and tedious logistics process for matsutake mushrooms, the temperature, relative humidity, gas composition, microorganisms and other environmental parameters in the storage environment may change at various stages of the logistics process, so the quality of the matsutake is easily affected, resulting in a low commercial value [4].

The Internet of Things (IoT), which describes the equipment of physical objects with sensors, actuators, computing logic and connectivity, has attracted much attention [5,6,7,8]. It can be applied to smart parking, smart city, smart environment, agriculture, health monitoring processes and so on [9]. Previous monitoring systems could only monitor temperature and relative humidity in the micro-environment, and could not provide a comprehensive characterization of the quality changes and other environmental parameters, nor establish the relationship between them [10]. Studies have shown that other environmental indicators, such as the gas composition and microorganisms, also have a great impact on changes in the quality of the matsutake [4,11]. Some quality indicators, such as the pH value, soluble solids content (SSC), hardness and color, have a direct impact on the shelf life of matsutake [12,13]. Therefore, it is necessary to study not only the changes of environmental parameters, but also the changes in the quality indicators, and to combine the two to analyze the characteristics of the quality changes in matsutake.

The post-harvest quality change of the matsutake is an inevitable process, and the application of preservation technology can slow down this change. Various existing types of packaging for fruit and vegetable preservation will slow down the aging of the matsutake quality and extend its shelf life to varying degrees. Different packaging methods have different relationships with quality. The most common packaging method is simple refrigeration preservation packaging, which is far from sufficient for long-distance transportation. The current post-harvest preservation techniques mainly consist of modified atmosphere packaging, coating film preservation, chemical inhibitors and so on [14,15,16,17]. They are widely used in fruit and vegetable preservation packaging, but the quality improvement differs [16,18]. Therefore, it is necessary to explore which kind of preservation packaging to use to improve the matsutake quality.

Based on previous studies, this study proposed a dynamic monitoring and quality management system for post-harvest matsutake, with different types of preservation packaging in the cold chain to extend the shelf life and maintain the market value. The system monitored the changes in the environmental parameters and quality data, and analyzed the quality characteristics and storage micro-environment data changes. Compared with the previous system, the matsutake quality changes could be evaluated more comprehensively. Additionally, this study explored the quality change characteristics of matsutake with different packaging methods to obtain the best packaging method to preserve, reduce the quality loss, extend the shelf life and improve the market value of the matsutake. The system provided a more refined technical means for matsutake preservation after the system evaluation, and a more scientific packaging method for matsutake storage and transportation.

## 2. Materials and Methods

### 2.1. Logistics Process and Quality Change Mechanism Analysis

To understand the matsutake quality changes in the cold chain, a matsutake logistics survey revealed that the main processes are: picking, pre-cooling and cleaning/grading, packaging and storage, cold-chain transportation and sales [19]. Figure 1 shows the specific Tibetan matsutake post-harvest logistics process. Mechanical damage and environmental temperature were the most important factors affecting matsutake quality, so it was necessary to control temperature, monitor parameters, choose appropriate packaging methods, avoid damage and extend the shelf life. The pre-cooling temperature, environment, relative humidity, packaging methods and transportation methods had different effects on the matsutake quality changes.

Different preservation methods controlled different factors affecting the matsutake’s quality, thereby inhibiting the changes in its physiological activities and achieving the purpose of preservation [20]. The quality change mechanisms with different packaging methods is shown in Figure 2. We analyzed the main physiological activities of the matsutake after picking and came up with the main factors affecting its quality. The matsutake is still an organic life-form after picking, and it continues to carry out physiological activities, affecting the environment. According to the analysis results, transpiration and respiration were found to be the main physiological activities. Due to the high temperature during the picking period, fresh matsutake undergo strong transpiration and lose water, affecting the relative humidity. After picking, the matsutake undergo vigorous aerobic respiration, absorbing O_2_ from the storage environment, consuming their own nutrients, and releasing H_2_O, CO_2_ gas and a large amount of respiratory heat, affecting the temperature and the gas environment. Under low oxygen or anaerobic conditions, the matsutake undergo anaerobic respiration, releasing alcohol, lactic acid and other substances that promote decay and deterioration. The matsutake release ethylene gas during the post-harvest maturation process, which promotes their maturation and opening, and rapidly affects their quality. Microorganisms in the storage micro-environment continue to grow and consume nutrients, accelerating the matsutake’s quality change.

According to the analysis of the post-harvest quality change mechanism, the main factors affecting the matsutake post-harvest quality are the temperature and relative humidity in the storage environment, O_2_, CO_2_ and ethylene gas concentrations in the micro-environment, and microorganisms. The physiological activities lead to moisture loss and microbial infection, thus affecting the sensory and physicochemical indicators and, consequently, the appearance, nutritive value, medicinal value and commodity value. The sensory indicators include the hardness (cap and rod), color and odor. The physicochemical indicators include the pH value, soluble solids content (SSC) and moisture content (MC). As time increases, the odor of the matsutake gradually fades, browning occurs on the surface and spoilage produces off-flavors and moldy spots. The hardness decreases with storage time because the matsutake rots and deteriorates before opening its umbrella.

Preservation packaging is one of the preservation technology tools that has a positive effect on the quality change of matsutake caused by the environmental changes and the shelf life. The current mainstream packaging method is simple refrigeration preservation packaging, which extends the shelf life by controlling the storage temperature in a relatively low range and reducing the cellular activity of the organism [17]. Modified atmosphere preservation adjusts the gas composition in the storage environment at a low temperature to create a low-temperature and low-oxygen storage environment, slowing down the metabolism and inhibiting the life activities of microorganisms, to extend the shelf life [21,22]. Hypobaric preservation is a non-polluting physical preservation technology using low-temperature storage, decompression treatment (or changing the air pressure of the matsutake storage environment) and controls gas composition to create a stable ultra-low oxygen environment to preserve matsutake [23]. With the continuous improvement of the preservation technology, irradiation preservation, coating preservation and preservative preservation technologies are also gradually applied to the preservation of edible mushrooms [24,25,26]. With the development of gas preservative preservation technology, compared with chemical preservative preservation technology, the gas preservative has the advantages of no pollution and no residue, simple operation and a good preservation effect, etc. SO_2_ gas preservative can inhibit and kill pathogenic microorganisms attached after picking, combined with refrigeration preservation technology to achieve the purpose of preservation.

### 2.2. System Architecture

To monitor the micro-environmental signals, a matsutake cold chain monitoring and tracing system was established. Figure 3a shows the overall structure of this system. The system consisted of three layers: the data acquisition layer, the data transmission and storage layer and the data application layer. The sensor module in the data acquisition layer was placed in the cold chain to sense the real-time parameters and send them to the information processing module for analysis. After the analog-to-digital conversion and parameter processing, the data transmission and storage layer received the processed data, sent the data to the data application layer for storage and communicated with the mobile terminal through the mobile communication switching center. The basic function of the data application layer is data collection, storage, evaluation and analysis. The data application layer consisted of a server layer and a client layer. The server layer connects the manager with the wireless information collection node, provides real-time information and alerts. The client layer provides the historical data and operational configuration interfaces for the matsutake micro-environment in the cold chain.

#### 2.2.1. Hardware Architecture Design

The design of the hardware system structure is shown in Figure 3b. This experiment adopted the matsutake cold chain real-time monitoring and tracing system. The hardware device consisted of three modules: a sensor module, an information processing module and a communication module. The sensor module consisted of a temperature and relative humidity sensor AM2322 (Ao Song, Guangzhou, China), a CO_2_ gas sensor, an O_2_ gas sensor (Analytical Technologies, New York, NY, USA) and an ethylene gas sensor. It sensed and collected environmental information in the cold chain through the sensors according to the user-defined interval. The STC12LE5A60S2 (STC micro TM, Shanghai, China) was used as the master control chip of the information processing module. The information processing module processes the analog and digital signals received from the sensor module, including converting the analog voltage signals of the gas sensor into digital signals through the analog-to-digital conversion module, and sorting and storing all the data information. The system used 433MHz as the wireless frequency to improve transmission performance and form the wireless sensor network (WSN). The system was based on the wireless sensor network and monitored the matsutake cold chain storage micro-environmental parameters in real time through the sensor information collection node device, transmitted the data to the information processing module through the interface, and transferred them to the background system database through the 4G communication module to storage. After specific data analysis by the computer program, the user can display information about the monitored parameters of the matsutake during the cold chain process, such as real-time data, location and so on.

#### 2.2.2. Software Architecture Design

The design of the software system structure is shown in Figure 3c. The software is developed based on the IoT’s TLINK cloud service platform, which is an extension of the connectivity tool that provides real-time connectivity to millions of sensing nodes and supports standard protocol integration for IoT such as MQTT (Message Queuing Telemetry Transport), SEP (Spanning Tree Protocol), etc. It supports open API (Application Programming Interface) interfaces for the secondary processing and development of data. The IoT-based TLINK cloud service platform application software allows users to monitor the cold chain environment in real time and query the historical data of the cold chain environmental information. The overall software structure includes the user center, monitoring center, and the evaluation, information feedback and exploration modules. According to the requirements of multi-parameter information collection and fusion, multi-sensor integration of the temperature, humidity, and gas (O_2_, CO_2_ and ethylene) was realized. Users set personal information and account passwords via the user center; then, data query and display, data analysis and data storage are run in the monitoring center module; after that, the evaluation module evaluates the quality, based on diagnosis rules, and the information feedback module notifies users and feeds back their treatment measures. Finally, users can search and add other IoT devices to the exploration module.

The monitoring system software was mainly responsible for the data analysis and display functions, as well as providing alarm functions. After it was connected to the sensing equipment of the multi-parameter information acquisition and fusion device, it formed a complete data transmission process. The specific workflow of the software was as follows:Step 1: Initialization. The sensor device node was matched and judged whether the connection was successful. After that, the software started to accept the data sent by the sensor device.Step 2: Data collection and analysis. The received data were analyzed and displayed by the monitoring center module. At the same time, the data were judged whether they exceeded the threshold, and the trigger would alert when they did.Step 3: Data storage. The monitoring center was determined to be connected or not to the sensing device; if it was connected, it would continue to receive and store data; otherwise, it would stop.

Figure 4 shows the interface of the matsutake cold chain real-time monitoring and tracing system, which displays real-time data and location information, etc., in the web browser.

### 2.3. Experiment Design

The matsutake samples were divided into four groups and used to measure the micro-environmental parameters changes in the cold chain and the quality indicator changes with different packaging methods (refrigeration preservation packaging, modified atmosphere packaging and preservative preservation packaging). Figure 5 shows some samples of matsutake. The matsutake cold chain micro-environmental parameters include the temperature, humidity and gas (O_2_, CO_2_ and ethylene). The quality indicators measured in the experiment were divided into sensory indicators and physicochemical indicators. The sensory indicators include the hardness (cap and rod), color and odor, and they were scored by the assessment team. The physicochemical indicators include the pH value, soluble solids content (SSC) and moisture content (MC). The pH value was obtained by measuring the filtered clear liquid with a pHB-4pH meter (INESA.CC, Shanghai, China). The filtered clear liquid was measured by a glycol refractometer (LH-B55, Hangzhou, Zhejiang, China) to obtain the content of the SSC. The SFY moisture meter (Guanya Electronic Technology Co., LTD, Shenzhen, China) was used to determine the MC of the samples.

PP (Polypropylene) has the advantages of a light weight, good mechanical properties and heat resistance, safety and health. PE (Polyethylene) has good air permeability, flexibility and chemical stability, good, low-temperature resistance and heat-sealing, and can be made into film and film bags for food packaging. The containers used for the experiments were hollowed out polypropylene (PP) baskets (length 23 cm × width 16 cm × height 9 cm) and polyethylene (PE) vacuum-packed cling bags (Weide New Material Co., Ltd., Xuzhou, Jiangsu, China) (length 20 cm × width 15 cm). The PP baskets were sealed with polyethylene (PE) cling film of 0.02 mm thickness. Each bag of matsutake packed in a cling bag was about 0.2 kg, and one bag was taken for each experiment. The temperature and relative humidity of the thermostatic incubator used for the experiments were 4 ± 1 °C and 90% RH (Relative Humidity), respectively. The physicochemical indicators of each experiment were measured three times and averaged.

#### 2.3.1. Micro-Environment Monitoring

The matsutake samples used in the experiment were divided into three groups, each group was 1 kg and each matsutake sample was separated by a special, absorbent paper for freshness preservation. The sensor probe connected to the cold chain sensing monitoring equipment and the sample used in the experiment were installed in the PP basket, sealed and placed in a cold store at 2–5 °C for testing. Data were recorded every 2 min and, when significant decay of the sample was observed, the monitoring was stopped; the entire monitoring test lasted 16 days.

#### 2.3.2. Different Preservation Packaging Experiment

**Refrigeration preservation packaging:** The matsutake samples were equally divided into two groups (T1 and T2), sealed in the PP baskets and put into the incubator at 0 °C and 4 °C. The T1 group underwent the experiment every 2 days and the T2 group underwent the test every 1.5 days. Each experiment treated two matsutake to determine the quality indicators.

**Modified atmosphere packaging:** The matsutake samples were recorded as the G3 group. According to Shuang Zhao’s previous experimental results, this experiment adopted a set of modified atmosphere ratios with the best preservation effect for fresh matsutake, and the specific gas compositions were as follows: the O_2_ concentration was 1%, the CO_2_ concentration was 21% and the N_2_ concentration was 78% [27]. The matsutake were pre-treated and packaged in the PE vacuum cling bags. A ZQF550/4 vacuum package machine and GM-B gas mixer (Justeel Machinery Manufacturing Co., LTD, Shanghai, China) were used together; the mixed gas was filled into the bags to replace the air and then sealed. The packaged matsutake were placed in the thermostatic incubator and the quality indicators were measured every 2 days.

**Preservative preservation packaging:** The matsutake samples were recorded as the P4 group. The matsutake was pre-treated and packaged in the PE vacuum cling bags, and the air mixed with 10 ppm SO_2_ gas was filled into the packages; they were placed in the incubator and the quality indicators measured every day.

### 2.4. Statistical Analysis

A parametric analysis was performed on independent samples from each treatment. The obtained data were analyzed by a double-factor analysis of variance (ANOVA) with SPSS software (International Business Machines Corporation, version 20, New York, NY, USA), to evaluate the impact of processing and storage time on the determined parameters. The Pearson correlation coefficient was used to evaluate the relationship between the various parameters, while Duncan’s multiple range test was used to identify the difference in the mean value of *p* ≤ 0.05.

## 3. Results

### 3.1. Matsutake Quality Indicators Changes at Different Packaging Methods

#### 3.1.1. Sensory Indicators Changes

The group of trained assessors included a local picker, an acquirer, a chef and two consumers. All the five assessors scored the color, hardness and odor of the matsutake. They observed and filled out the form every day and recorded the scores. In this study, the sensory assessment is divided into four levels, and the specific assessment criteria are shown in Table 1. In the processing of the sensory assessment data, the final sensory score was the median of the assessment value.

The four sensory evaluation indicator changes with the different packaging methods are shown in Figure 6. The evaluation value of the four sensory indicators of the matsutake decreased with time, and the odor, which is a sensitive indicator, decreased the fastest. The cap hardness in the 4 °C refrigerated preservation packaging experiment was lower, compared to the remaining three groups, and the rod hardness in the modified atmosphere packaging experiment decreased slowly and was relatively high overall. The sensory indicators of the matsutake remained good in the first 4 days and decreased faster from day 5 to day 10, and the individual variability of the matsutake became greater after day 10. The shelf life of the matsutake under the four experimental conditions was 12 days, 10.5 days, 18 days and 9 days, respectively. In the refrigerated preservation packaging experiments, it was found that, although the shelf life of matsutake was longer at 0 °C, frost damage occurred. Considering the losses and refrigeration costs, it is more appropriate to store the matsutake in an incubator at 4 °C under the conditions of this experiment.

#### 3.1.2. Physicochemical Indicator Changes

Figure 7 shows the characteristics chart of the matsutake physicochemical indicator changes with the different packaging methods.

The matsutake physicochemical indicator changes with the refrigeration preservation packaging at different temperatures are shown in Figure 7a. During the storage time, the pH value increased and then decreased with time; the SSC decreased and then increased with time, while the MC showed a decreasing trend. After the storage time, the changes showed greater individual variability, with the pH value showing an overall decreasing trend; the SSC showing an overall decreasing trend followed by an increasing trend, and the MC showing an overall decreasing trend. Under the conditions of this experiment, the pH value of the matsutake was maintained between 6.66 and 6.88, with a variation interval of about 0.2; the SSC was maintained between 0.88% and 1.29%, with a variation interval of about 0.40; and the MC was maintained above 84.8% during storage. In the early storage stage, the matsutake consumed their own nutrients for physiological activities, so the SSC decreased. However, as the MC decreased, the SSC concentration increased. Therefore, in the late storage stage, the SSC and MC showed a certain correlation.

Figure 7b shows the matsutake physicochemical indicator changes with the modified atmosphere packaging, which can be divided into three stages: S1, S2 and S3. In the S1 stage, the pH value increased rapidly; the SSC and MC decreased. In the S2 stage, the pH value and the MC decreased slowly, and the SSC remained stable. In the S3 stage, the pH value, SSC and MC fluctuated dramatically. With the increase in time, the matsutake’s individual quality differences increased significantly, and there was a large data fluctuation. Comparing the SSC and MC changes in the S3 stage, it was found that, when the MC decreased, the SSC increased between the 4th and 10th day; when the MC increased, the SSC decreased between the 14th and 18th day. The results indicate that there is a correlation between the SSC and the MC during the fresh matsutake storage stage. Through sensory evaluation, we concluded that the matsutake’s shelf life was terminated between days 14 and 18, a period with high volatility in the pH value, SSC and MC changes, which was defined as the decay stage.

Figure 7c shows the matsutake physicochemical indicator changes with the preservative preservation packaging, which can be divided into three stages: S1, S2 and S3. In the S1 stage, the pH value increased, and the SSC and MC decreased. In the S2 stage, the pH value and the MC decreased; the SSC increased. In the S3 stage, the matsutake individual quality differences increased significantly, and there was a large data fluctuation. In the S2 stage, the continued decrease in MC resulted in a relative increase in SSC, and the MC was linearly negatively correlated with the SSC. It can be observed that, under the conditions of this experiment, the matsutake quality can be assured in the early preservation stage (0–5 days); after the fifth day, the matsutake quality decreased rapidly. Under the conditions of this experiment, the matsutake pH value was maintained between 6.67 and 6.9, with a variation interval of about 0.2, during the shelf life. The SSC was maintained between 1.00% and 1.50%, with a variation interval of about 0.50. The MC was maintained above 85.9%.

Based on the above analysis of the matsutake physicochemical indicator changes with the different packaging methods, the matsutake physiological change process was divided into three stages: S1, S2 and S3. The S1 stage is the continuous growth stage: fresh matsutake are physiologically active after harvesting and consume nutrients towards maturity. The S2 stage is the quality stability stage: the matsutake quality varies very steadily. The S3 stage is the spoilage stage: the matsutake shows decay and deterioration, and the quality is highly variable and fluctuates individually.

### 3.2. Changes in Matsutake Micro-Environmental Parameters

In this experiment, the matsutake micro-environmental parameters in the cold chain were characterized at 2–5 °C. The temperature and relative humidity changes and gas environment changes in the matsutake storage micro-environment were measured. The sensors used for measuring the temperature and relative humidity, O_2_, CO_2_ and ethylene were SHT11, SHT11, AJD-4M-O_2_, AJD/L/4CO_2_ and A15-75D, respectively. Due to the large sample size and the data reproducibility obtained in the experiment, the moving average method was selected to smooth the obtained data, according to one step for every ten data units when processing the data, and the data unit was 10 intervals. The whole monitoring process can be divided into three stages: S1, S2 and S3.

The changes in temperature and relative humidity in the matsutake micro-environment are presented in Figure 8a. In the S1 stage, the temperature and relative humidity decreased dramatically. In the S2 stage, the temperature and relative humidity were in a relatively stable state. In the S3 stage, the temperature and relative humidity began to decrease gradually.

Figure 8b shows the changes in the gas concentration in the matsutake micro-environment. In the S1 stage, the gas concentration in the matsutake micro-environment changed rapidly, with the O_2_ concentration decreasing and the CO_2_ and ethylene concentrations increasing. In the S2 stage, the ethylene concentration reached its maximum; the O_2_ concentration decreased slowly at a lower rate than that in the S1 stage, and the CO_2_ concentration increased slowly. In the S3 stage, the O_2_ concentration continued to decrease and the rate became slower; the CO_2_ concentration rose slowly, and the ethylene concentration decreased. In the whole process, the ethylene concentration was low and the measurement accuracy was high, so the volatility was large. The variation range was 0–2 ppm, which was within the error range. Due to fluctuations in stability during the cold chain transportation, vibration can cause a significant increase in the measured gas concentration. The measured gas concentration will decrease significantly, due to the decrease in airtightness resulting from the package leakage.

### 3.3. Data Analysis

According to the correlation results of each indicator in Figure 9, it is possible to determine the relationship between the quality indicators and the matsutake’s remaining shelf life. The SSC and remaining shelf life showed a negative correlation, whereas the other quality indicators and remaining shelf life showed a positive correlation. The correlation coefficients between pH and remaining shelf life and SSC and remaining shelf life were significantly different and less correlated under different packaging methods; this was due to the phase effect of matsutake quality changes. Therefore, the pH value and SSC were excluded in the matsutake quality prediction model. The correlation coefficient between the cap hardness and the remaining shelf life ranged from 0.80 to 0.96, and the correlation coefficient between the color and the remaining shelf life ranged from 0.84 to 0.98, both of which were relatively weak. As a result, the cap hardness and color also were excluded in the matsutake quality prediction model. The correlation coefficients of odor and remaining shelf life under different packaging methods were 0.95, 0.95, 0.96 and 0.98, all above 0.95; the correlation degree was obvious, and the correlation coefficients of rod hardness and remaining shelf life and MC and remaining shelf life under different packaging methods were above 0.85, so rod hardness, odor and MC showed a strong linear correlation with remaining shelf life.

Figure 10 shows the cluster analysis of the matsutake hardness (cap and rod), color, odor, pH, SSC and MC with the different packaging methods. They were divided into different clusters, so that the objects in the same cluster have great similarity, and the objects in the different clusters have great differences. Supposing the two n-dimensional data points are A = (*a*_1_, *a*_2_..., *a_n_*) and B = (*b*_1_, *b*_2_..., *b_n_*), then the Euclidean distance between A and B is:(1)$$\rho (A,\;B)=\sqrt{\sum_{i=1}^{n}{\left(a_{i}-b_{i}\right)}^{2}},\;i=1,2,\ldots ,n$$

The red ellipse represents a 95% confidence interval, which verifies the feasibility of the cluster analysis method. In this paper, the quality indicators of the matsutake were divided into two categories, namely, two cluster centroids were obtained, and finally two comprehensive indicators were obtained: cluster 1(PC1) and cluster 2(PC2); PC1 and PC2 contained most of the information of the original data. It is easy to see that the SSC has a strong positive correlation with PC2, the MC and pH have a negative correlation with PC2. The hardness (cap, rod), color and odor have a strong positive correlation with PC1. Therefore, the seven quality indicators were divided into physicochemical categories to represent the matsutake shelf life. The physical indicators, including the rod hardness, color and so on, accounted for the most important proportion (~70%), while the chemical indicators, including the SSC, pH and so on, had less influence (~20%).

### 3.4. System Evaluation

In this study, the relevant personnel were invited to participate in the system performance analysis and offer optimization suggestions, which improves the applicability of the established monitoring system. The evaluation team of the system was composed of staff, managers, research group members and experts from the cold chain logistics-related enterprises. They participated in the evaluation and discussion of the system to evaluate whether the function and the performance of the system could meet the requirements and whether it could be further improved. The optimized model could provide a more effective theoretical reference for the dynamic monitoring and real-time quality management traceability of the monitoring and management system. Through the data analysis of the collected data, the established matsutake preservation packaging and quality management system can evaluate the different preservation packaging methods well. The performance metrics are listed in Table 2. The sensory evaluation indicator values and the consumer acceptance threshold were used as indicators to determine the termination of the matsutake’s shelf life, and the remaining shelf life was used to characterize the matsutake’s quality changes. Therefore, the total matsutake shelf life with the different packaging methods was used as an indicator to determine their preservation effect, and the results comparison of the matsutake preservation effects in the cold chain with the different packaging methods is shown in Table 2, namely, refrigeration preservation packaging, modified atmosphere packaging and preservative preservation packaging.

## 4. Conclusions

In this paper, we analyzed the logistics process for fresh matsutake after picking and the mechanisms of quality change, and we derived the quality-influencing factors of the matsutake. For the post-harvest cold chain transportation of matsutake, a dynamic monitoring and quality management system under different preservation packaging in the cold chain was designed and optimized, adding O_2_, CO_2_ and ethylene sensors to the traditional IoT, and building the monitoring software of the remote monitoring center based on the TLINK platform to make the applicability of the system comprehensive, real-time, online and simple. The system monitored the micro-environmental parameters such as temperature, relative humidity, O_2_, CO_2_ and ethylene during the cold chain transportation of the matsutake in real time, analyzed the data change pattern and the quality indicators, visualized the cold chain logistics process, and provided a more refined technical means for matsutake preservation.

The cold chain micro-environmental monitoring experiment and matsutake preservation packaging experiment, based on three different preservation methods (refrigeration preservation packaging, modified atmosphere packaging and preservative preservation packaging), were designed. We analyzed the changes in the storage micro-environment data and quality change characteristics of the matsutake after picking, and we explored the effect of different preservation methods in the cold chain on the quality of the matsutake. In conclusion, the matsutake shelf life was relatively long and best preserved with the modified atmosphere packaging method (1% O_2_, 21% CO_2_ and 78% N_2_), with a shelf life of 18 days, and the system was verified. Through correlation analysis and cluster analysis, it was concluded that the three indicators of rod hardness, odor and MC were important for the research into the remaining shelf life, and the method was provided for the preservation of the fresh matsutake in the cold chain process to extend its shelf life and improve its market value.

## Figures and Tables

**Figure 1 foods-11-02646-f001:**
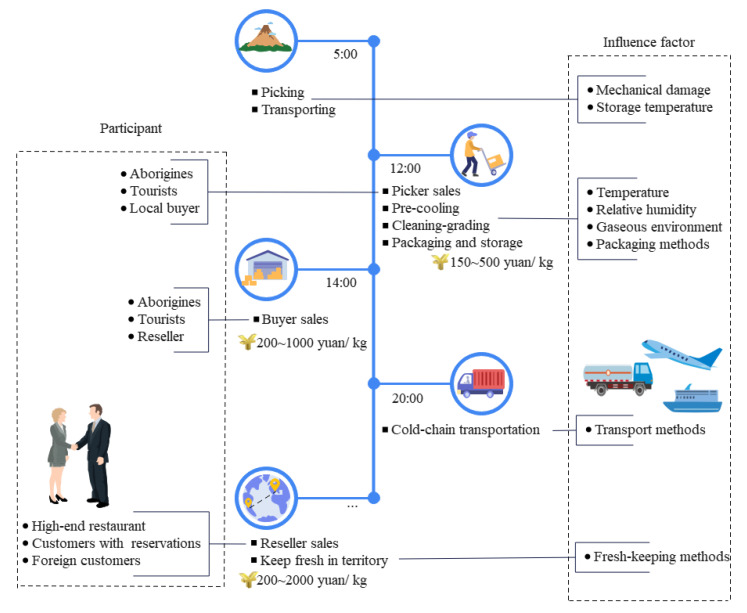
Tibetan matsutake post-harvest logistics flow chart.

**Figure 2 foods-11-02646-f002:**
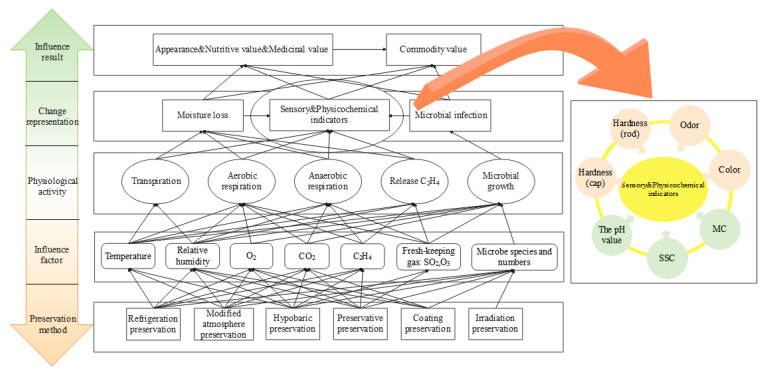
Matsutake quality change mechanism during the preservation process.

**Figure 3 foods-11-02646-f003:**
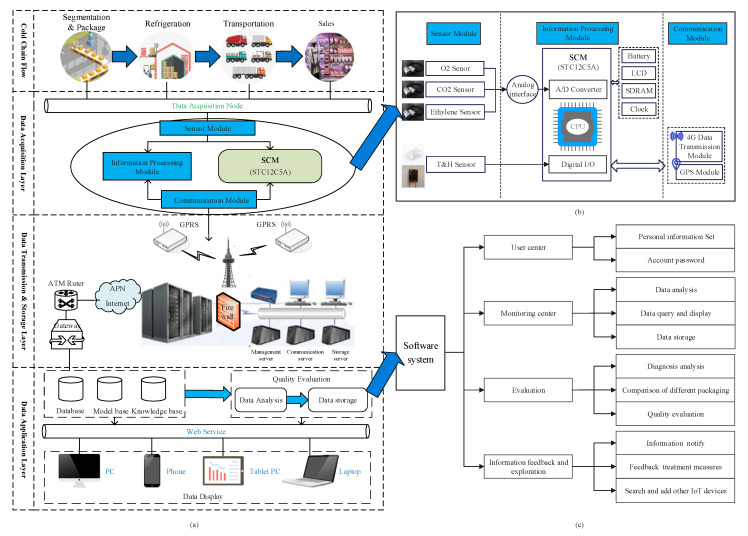
(**a**) The overall structure of the matsutake cold chain monitoring and tracing system; (**b**) the design of the hardware system structure; (**c**) the design of the software system structure.

**Figure 4 foods-11-02646-f004:**
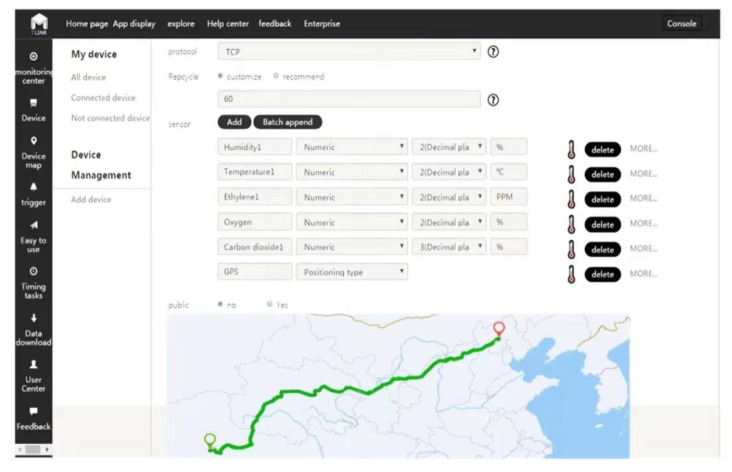
The interface of the matsutake cold chain real-time monitoring and tracing system.

**Figure 5 foods-11-02646-f005:**
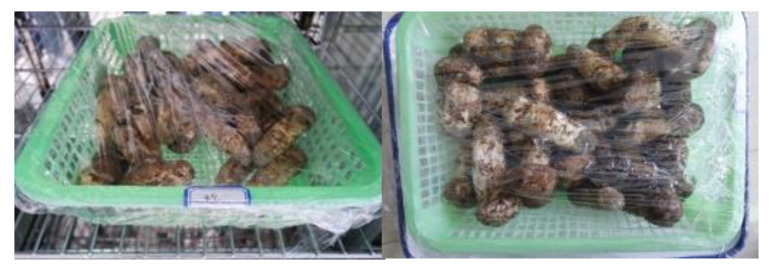
Display of matsutake samples.

**Figure 6 foods-11-02646-f006:**
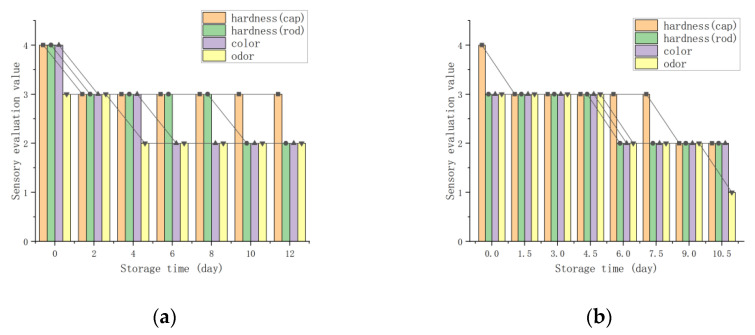
Matsutake sensory evaluation indicator changes with the different packaging methods: (**a**) refrigeration preservation packaging at 0 °C; (**b**) refrigeration preservation packaging at 4 °C; (**c**) modified atmosphere packaging; (**d**) preservative preservation packaging.

**Figure 7 foods-11-02646-f007:**
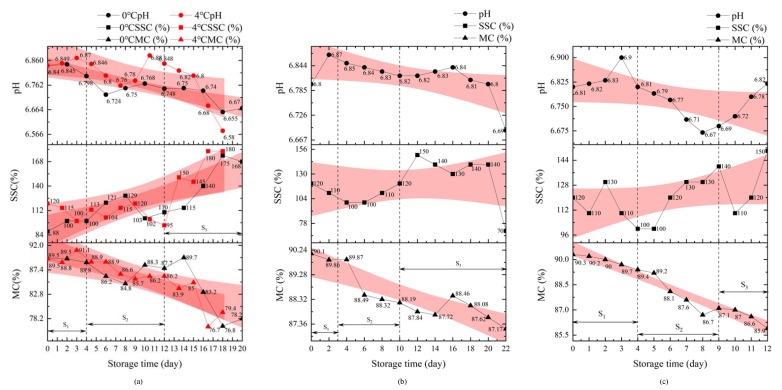
Characteristics chart of matsutake physicochemical indicator changes with the different packaging methods: (**a**) refrigeration preservation packaging at different temperatures; (**b**) modified atmosphere packaging; (**c**) preservative preservation packaging.

**Figure 8 foods-11-02646-f008:**
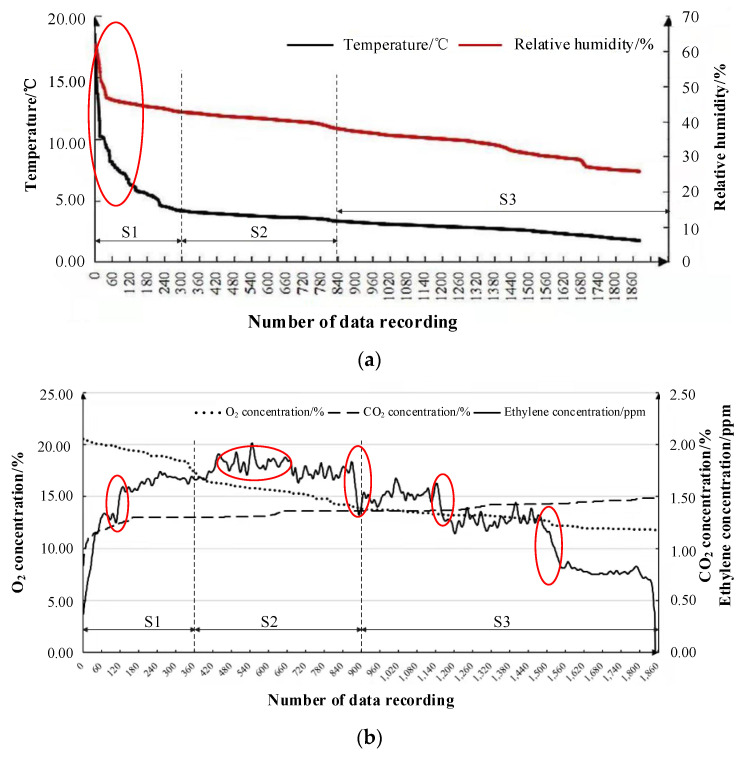
Matsutake micro-environmental parameter changes during at 2–5 °C: (**a**) scatter plot of temperature and relative humidity changes; (**b**) gas concentration changes.

**Figure 9 foods-11-02646-f009:**
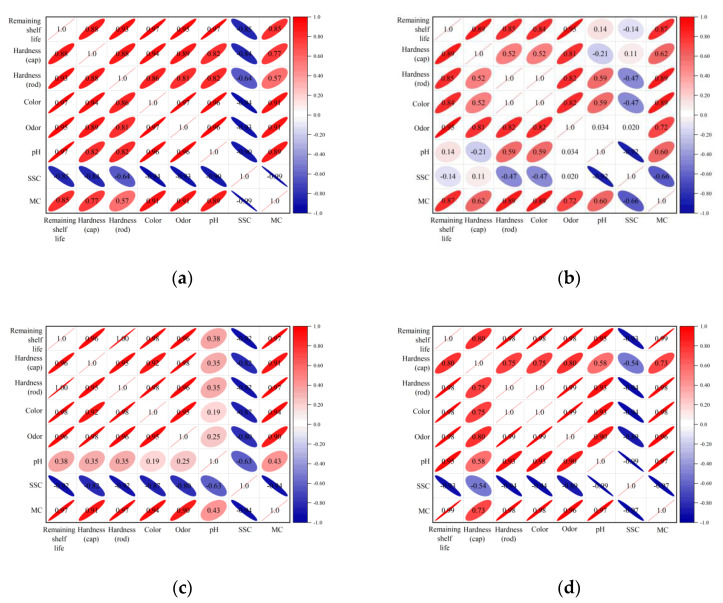
Correlation coefficients between matsutake quality indicators and remaining shelf life: (**a**) refrigeration preservation packaging at 0 °C; (**b**) refrigeration preservation packaging at 4 °C; (**c**) modified atmosphere packaging; (**d**) preservative preservation packaging.

**Figure 10 foods-11-02646-f010:**
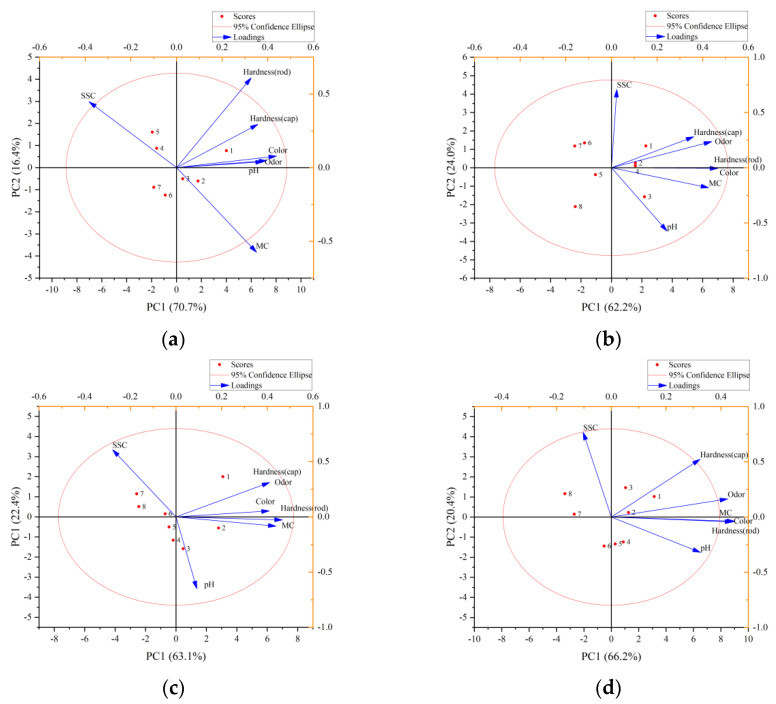
Cluster analysis: (**a**) refrigeration preservation packaging at 0 °C; (**b**) refrigeration preservation packaging at 4 °C; (**c**) modified atmosphere packaging; (**d**) preservative preservation packaging.

**Table 1 foods-11-02646-t001:** Matsutake sensory indicator score table.

Score	Color	Hardness	Odor
4	The cap is white, and the rod is milky white without browning	The cap is elastic and the rod is hard	Fresh smell
3	Normal in color with mild browning	The cap has better elasticity and the rod is stiffer	Normal
2	Moderate browning with dark caps	The cap and rod begin to soften	A slight odor
1	Severe browning with mildew	Severe softening, water seepage	A strong odor

**Table 2 foods-11-02646-t002:** System performance comparison.

System	Parameters	Type	Resolution	Range	Accuracy	Response Time	Packaging Method	Group	Shelf Life	Applicability	Other
Previous system	Temperature	SHT11	0.1	−40 °C~120 °C	±0.4 °C	-	-	-	-	Easy, simple	-
	Humidity	SHT11	0.1	0~100% RH	±3% RH	-	-	-	-		
The cold chain real-time monitoring and tracing system	Temperature	SHT11	0.1	−40 °C~80 °C	±0.3 °C	-	Refrigeration preservation packaging at different temperature	1	12.0 d	Comprehensive, real-time, online and simple	Add real-time alerts
	Humidity	SHT11	0.1	0~100% RH	±2% RH	-		2	10.5 d		
	O_2_	AJD-4M-O_2_	0.1	0~30% vol	±1% FC	<25 s	Modified atmosphere packaging (4 °C, 90% RH)	3	18.0 d		
	CO_2_	AJD/L/4CO_2_	0.1	0~5% vol	±2% FC	<25 s	Preservative preservation packaging (4 °C, 90% RH)	4	9.0 d		
	C_2_H_4_	A15-75D	0.1	0 ppm~100 ppm	±1 ppm	<25 s	-	-	-		
Advantage	Increased monitoring parameters, the monitoring and traceability of CO_2_, O_2_ and ethylene, better accuracy and traceability of temperature and humidity, improving the management of the cold chain										

## Data Availability

The data presented in this study are available on request from the corresponding author.

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
