# Peer review of "Optimized Dynamic Monitoring and Quality Management System for Post-Harvest Matsutake of Different Preservation Packaging in Cold Chain"

_foods, 2022, doi:10.3390/foods11172646_

Round 1

Reviewer 1 Report

The article is very well written. However, some changes are required, which are mentioned below.

Line 14: Start the sentence with 'The', instead of 'And'.

Line 41: Check the grammar of the sentence starting from line 41. 

Line 45: Some repetition of words. 

Line 47: First write what is meant by SSC. 

64: Better to write the application name (TLINK) in the Methodology section, not in the Introduction. 

72: The processes can be written without '&'.

79: The need of the study should only be provided in the Introduction. 

81: Remove 'AM' from the time mentioned in the figure. 

138: Red and blue colors are less suitable with black colored text. 

142: Join sentence ending at line 142 with line 143 to make one paragraph. 

144: Replace '&' with 'and'. Do the same for rest of the document. 

146: Describe SCM and A/D.

165: Do not start sentence with 'And'. Do the same for rest of the document. 

221: Describe more about the assessment team. 

226: Company name should be properly mentioned. 

229: Spaces should be provided before 'x'.

248: Delete parenthesis for 'PP'.

265: Provide space after '(ANOVA)'.

278: Change 'Modified' to 'modified'.

Some general comments about Introduction:

- There is need to cite the work already done and to clearly mention the research gaps related to the present study.

- The novelty of the work should be highlighted.  

- Some of the material can be shifted from 'Materials and Methods' to 'Introduction. 

Some general comments about 'Materials and Methods':

- Should be shortened and more specific.

- Provide required references. There are certain occasions where there is need to provide proper references. 

- System evaluation should be provided. Explain 'relevant personnel'.

Reviewer 2 Report

Do you have data showing the value of losing to matsutake after harvest? In the china and  global perspective  ? it will better if can shows in the introduction section

Why did you conclude that the matsutake were best preserved under the gas conditions of 1%O2, 21% CO2 and 78% N2 is the best tratment ?

Why you did not measure The microbiology parameter since this product is fungi product ?

Line : 18 : 1%O2 need a space

Line 19 : And data analysis : it will be better if word “ And” not after dot.

Line 64 : What is TLINK ?

Line  221 : How many panelists did you use in this experiment?

Line 228-229 : Why did you use  PP and PE as pacakging material ?  

Line 255-256 : How did you achieve the gas composition?

Line 260 : Which kind type of plastic bag did you use?

Line 261 : 10ppm => need a space ?

Line 274-286 : the best treatment for sensory evaluation should be more elaborated

Line 286 : What the meaning of number In  y-axis?

Line 334 : ( d) Preservative preservation packaging è no figure ???? How can you clarify?

Line 291-330 : The explanation is unconnected with the figure. For example : line 307 about figure 7(b) explain about MAP but in figure is written 7(c). Which is one correct?  

Line 355-366 : What type of sensors did you use to measure Co2, O2, N and ethylene ? it should be mentioned in method

Line 1 : Conclusion should be easily readable………if needed it could be written in different paragraphs

Round 2

Reviewer 2 Report

Thank you for your email, I think the authors have revised their paper correctly.   Best regards